# *XPO1* Expression Is a Poor-Prognosis Marker in Pancreatic Adenocarcinoma

**DOI:** 10.3390/jcm8050596

**Published:** 2019-04-30

**Authors:** David Jérémie Birnbaum, Pascal Finetti, Daniel Birnbaum, Emilie Mamessier, François Bertucci

**Affiliations:** 1Laboratoire d’Oncologie Prédictive, Centre de Recherche en Cancérologie de Marseille, Aix-Marseille Université, INSERM UMR1068, CNRS UMR725, F-13273 Marseille, France; david.birnbaum10@gmail.com (D.J.B.); finettip@ipc.unicancer.fr (P.F.); daniel.birnbaum@inserm.fr (D.B.); emilie.mamessier@inserm.fr (E.M.); 2Département d’Oncologie Médicale, Institut Paoli-Calmettes, F-13273 Marseille, France; 3Département de Chirurgie Générale et Viscérale, AP-HM, F-13000 Marseille, France

**Keywords:** pancreas cancer, prognosis, survival, *XPO1* expression

## Abstract

Pancreatic adenocarcinoma (PAC) is one of the most aggressive human cancers and new systemic therapies are urgently needed. Exportin-1 (XPO1), which is a member of the importin-β superfamily of karyopherins, is the major exporter of many tumor suppressor proteins that are involved in the progression of PAC. Promising pre-clinical data using XPO1 inhibitors have been reported in PAC, but very few data are available regarding XPO1 expression in clinical samples. Retrospectively, we analyzed *XPO1* mRNA expression in 741 pancreatic samples, including 95 normal, 73 metastatic and 573 primary cancers samples, and searched for correlations with clinicopathological and molecular data, including overall survival. *XPO1* expression was heterogeneous across the samples, higher in metastatic samples than in the primary tumors, and higher in primaries than in the normal samples. “*XPO1*-high” tumors were associated with positive pathological lymph node status and aggressive molecular subtypes. They were also associated with shorter overall survival in both uni- and multivariate analyses. Supervised analysis between the “*XPO1*-high” and “*XPO1*-low” tumors identified a robust 268-gene signature, whereby ontology analysis suggested increased XPO1 activity in the “XPO1-high” tumors. *XPO1* expression refines the prognostication in PAC and higher expression exists in secondary versus primary tumors, which supports the development of *XPO1* inhibitors in this so-lethal disease.

## 1. Introduction

Pancreatic adenocarcinoma (PAC) is a major public health problem worldwide with the highest mortality rate of all human cancers and a rising incidence [1,2]. Complete surgical tumor resection followed by chemotherapy is the only curative treatment available, but less than 20% of patients are eligible for surgery at diagnosis [3,4]. In the case of inoperable or metastatic form, the median survival is six months and the long-term survival is null. The improvements in radiotherapy and systemic treatments during the past 20 years have achieved limited impact. The few chemotherapeutic agents that are efficient against PAC include gemcitabine with or without nab-paclitaxel and the FOLFIRINOX regimen that combines 5-FU, leucovorin, oxaliplatin, and irinotecan. The survival benefit is modest, making the development of novel drugs crucial. Few molecular alterations, such as *KRAS*, *TP53*, *SMAD4*, *CDKN2A*, *BRCA2*, and *ARID1A* mutations and *GATA6* amplification have been identified in PAC [5,6,7,8,9,10,11,12], but most of them remain non-druggable. Furthermore, many other tumor suppressors are not mutated, but inactivated through different post-translational mechanisms. Thus, PACs are very heterogeneous and they display alterations in many critical biological pathways, making the design of therapy against a single pathway unrealistic. A therapy that can simultaneously target the inactivation of multiple tumor suppressor proteins (TSPs) is worthy of investigation.

The active transport of macromolecules between the nucleus and the cytoplasm controls the localization and functions of many proteins and RNAs. Exportin-1, which is also called chromosome region maintenance 1 (CRM1/XPO1), is a member of the importin-β superfamily of karyopherins; it is the major exporter of the majority of proteins, some mRNAs, rRNAs, and snRNAs [13], from the cell nucleus to the cytoplasm. Currently, there are more than 200 known export targets of XPO1 that are involved in various cellular functions and diseases. In eukaryotic cells, XPO1 is considered to be the sole exporter of most of the tumor suppressor proteins (TSPs), such as p53, BRCA1/2, survivin, nucleophosmin, APC, and FOXO, leading to their inactivation through mislocalization [14,15]. Thus, many TSPs are inactivated in the case of increased XPO1 expression and they cannot be activated, even in the presence of chemotherapeutics or targeted drugs. Exportin 1 is also involved in the activation of oncogenic pathways through the enhanced nuclear export of EIF4E, which is the sole transporter of guanine-capped mRNAs, including mRNAs for oncogenes, such as MYC, cyclin D1, and MDM2. XPO1 exports also microRNAs in addition to its nuclear protein export function [16]. Thus, high XPO1 activity favors tumor progression and therapeutic resistance. High expression levels have been found in many types of solid and hematological malignant tumors, including cervical, prostate, ovarian, and gastric cancers, osteosarcoma, glioma, multiple myeloma, lymphoma, and leukemia, and they were often associated with unfavorable clinical outcomes [17,18,19,20,21,22,23,24,25,26,27].

XPO1 inhibition can inhibit several critical pathways that promote cancer progression and therapy resistance, and became a potential therapeutic strategy. This has led to the development of XPO1 inhibitors, initially including natural products, such as leptomycin B [28,29], plumbagin [17], and curcumin [30], and then chemically synthesized inhibitors [31,32,33]. More recently, the selective inhibitor of nuclear export (SINE) compounds, a unique class of XPO1 inhibitors [34,35,36,37], were developed. These compounds showed low nanomolar IC50s against cancer cells and they have normal cell sparing properties. Their anti-tumor activity, either alone or in combination with respective standard drugs, was documented in many solid cancers and hematological malignancies [35,38,39,40,41,42,43,44,45,46]. Currently, the dominant XPO1 inhibitor, selinexor (KPT-330) and its related analog altenexor, are being assessed in several phase I-II clinical trials for different cancers.

In PAC, preclinical data support the development of XPO1 inhibitors [47,48,49,50,51]. For example, SINE blocked pancreatic cancer cell proliferation, induced apoptosis, and retained important TSPs in the nucleus [47]. The association of SINE with gemcitabine-nab-paclitaxel drastically reduced the growth of pancreatic cancer cell lines, suppressed spheroid formation in CSCs, and blocked CSC xenograft growth. Furthermore, XPO1 inhibition by selinexor increased miR-145 expression in pancreatic cancer cells, resulting in decreased cell proliferation and migratory capacities [50]. Clinical trials are ongoing, such as the NCT02178436 phase Ib/II clinical study.

However, there is a paucity of data regarding the prevalence of XPO1 expression in clinical samples of pancreatic cancer, with only three studies [23,52,53], only including one that analyzed correlations with clinicopathological features and survival in a cohort of 76 patients [52]. Here, we have analyzed *XPO1* mRNA expression in a clinical series of 741 pancreatic samples, including 95 normal samples, 73 metastatic samples of PAC, and 573 primary PAC, and searched for correlations with clinicopathological and molecular data, including overall survival.

## 2. Materials and Methods

### 2.1. Gene Expression Data Sets

We gathered clinicopathological and gene expression data of clinical pancreatic carcinoma samples from ten publicly available data sets [54,55,56,57,58,59,60,61,62,63], which comprise at least one probe set representing *XPO1* (Appendix A). Data were collected from the National Center for Biotechnology Information (NCBI)/Genbank GEO, ArrayExpress, and TCGA databases. The samples were profiled using whole-genome DNA microarrays (Affymetrix, Agilent) and RNASeq (Illumina). The pooled data set contained 1052 samples, including 573 primary PAC samples, 73 metastatic samples, and 95 normal pancreatic samples. Our institutional board approved the study. A total of 573 PAC samples that were informative for overall survival were included in the present analysis.

### 2.2. Gene Expression Data Analysis

Data analysis required pre-analytic processing. First, we separately normalized each DNA microarray-based data set by using quantile normalization for the available processed Agilent data, and Robust Multichip Average (RMA) with the non-parametric quantile algorithm for the raw Affymetrix data sets. Normalization was done in R using Bioconductor and the associated packages. Subsequently, we mapped hybridization probes across the different technological platforms present. We used SOURCE (http://smd.stanford.edu/cgi-bin/source/sourceSearch) and EntrezGene (Homo sapiens gene information db, release from 04/27/2017), ftp://ftp.ncbi.nlm.nih.gov/gene/) to retrieve and update the Agilent annotations, and NetAffx Annotation files (www.affymetrix.com; release from 01/12/2008) for the Affymetrix annotations. The probes were then mapped according to their EntrezGeneID and, when multiple probes represented the same GeneID, we retained the one with the highest variance in a particular dataset. For the RNA-seq data, we used the available normalized RNASeq data that we log_2_-transformed. Subsequently, we corrected the ten studies for batch effects using *z*-score normalization. Briefly, for each separate *XPO1* expression value in each study, subtracting the mean of the gene in that dataset divided by its standard deviation, mean, and standard deviation only being measured on primary cancer samples transformed the value. *XPO1* expression in tumors was measured as discrete value after comparison with mean expression in the 573 primary tumors: high expression was defined by value > mean and low expression by value ≤ mean.

We separately applied different multigene classifiers to each sample in each data set: the subtype classifiers that were reported by Bailey [61], Collisson [58], and Moffitt [60], and the 25-gene prognostic signature that we recently developed [64]. Finally, to explore the biological pathways linked to *XPO1* expression in pancreatic cancer more-in-depth, we applied a supervised analysis by using the largest data set (TCGA: 150 samples) as a learning set, and the other data sets as independent validation sets (423 samples). In the learning set, we compared the whole-genome expression profiles between tumors with (*N* = 76) versus without (*N* = 74) high *XPO1* expression using a moderated t-test with an empirical Bayes statistic [65] being included in the limma R packages. False discovery rate (FDR) [66] was applied to correct the multiple-testing hypothesis and the following thresholds: *p* < 1.0 × 10^−5^ and *q* < 1.0 × 10^−5^ defined significant genes. Ontology analysis of the resulting gene list was based on the GO biological processes of the Database for Annotation, Visualization and Integrated Discovery (DAVID; david.abcc.ncifcrf.gov/). We verified the robustness of the resulting gene list in the validation set (295 tumors with and 283 without high *XPO1* expression) by computing a metagene-based prediction score defined by the difference between the “metagene *XPO1*-high” (mean expression of all genes upregulated in the “*XPO1*-high” class) and the “metagene *XPO1*-low” (mean expression of all genes upregulated in the “*XPO1*-low” class) for each tumor. This score was then compared between the “*XPO1*-high” and “*XPO1*-low” samples.

### 2.3. Statistical Analysis

The *t*-test or the Fisher’s exact test, when appropriate, were used to analyze the correlations between *XPO1*-based tumor classes and clinicopathological features. Overall survival (OS) was calculated from the date of diagnosis to the date of death from pancreatic cancer. Follow-up was measured from the date of diagnosis to the date of last news for event-free patients. The survivals were calculated using the Kaplan–Meier method and the curves were compared with the log-rank test. Univariate and multivariate survival analyses were done using Cox regression analysis (Wald test). The variables tested in univariate analyses included patients’ age at time of diagnosis (continuous value), sex, American Joint Committee on Cancer (AJCC) stage (4, 3, 2 vs. 1), pathological features including pathological type, tumor grade (3, 2 vs. 1), tumor size (T4, T2, T3 vs. T1), regional lymph node status (positive vs. negative), and *XPO1* expression (“high” vs. “low”). Variables with a *p*-value < 0.05 were tested in multivariate analysis. All of the statistical tests were two-sided at the 5% level of significance. Statistical analysis was done using the survival package (version 2.30) in the R software (version 2.15.2; http://www.cran.r-project.org/). We followed the reporting REcommendations for tumor MARKer prognostic studies (REMARK criteria) [67].

## 3. Results

### 3.1. Patients’ Population

Table 1 summarizes the analyzed *XPO1* mRNA expression in 573 clinical primary PAC samples. Their clinicopathological characteristics are summarized. Briefly, most of patients were more than 60 year-old and 53% were male. Most of the tumors were ductal type (93%), grade 2 (57%), and they were classified as AJCC stage II (85%); most of them were pT3 tumors (78%) and most had at least one lymph node involved (70%). All but one had been initially treated by surgery. None of them had received neoadjuvant chemotherapy or radiotherapy. All of the molecular subtypes were represented with more frequent squamous Bailey’s subtype (36%), more frequent classical Collisson’s subtype (45%), more frequent classical Moffitt’s tumor subtype (60%), and more frequent activated Moffitt’s stroma subtype (59%).

### 3.2. XPO1 Expression and Clinicopathological Features

The *XPO1* expression was variable and different between the normal tissue samples, the primary tumors, and the metastatic samples (Figure 1A), with an increasing gradient from normal samples to primary cancer samples (*p* = 4.88 × 10^−18^, Student *t*-test), and from primary cancer samples to metastatic samples (*p* = 1.22 × 10^−21^, Student *t*-test). We defined two classes of primary cancer samples that were based upon XPO1 expression in tumors when compared with mean expression in normal pancreatic samples: the “*XPO1*-high” class (*N* = 298; 52%) and the “*XPO*-low” class (*N* = 275, 48%). We then searched for correlations between these two classes and the clinicopathological and molecular features (Table 1). There was no correlation with patient’s age and sex, AJCC stage, and pathological type, grade, and tumor size. By contrast, correlations (Fisher’s exact test) existed with the pathological lymph node status (pN) and the molecular subtypes (Figure 1B–E). The “*XPO1*-high” tumors were enriched in pN-positive tumors (*p* = 1.97 × 10^−2^) and in squamous Bailey’s subtype (*p* = 3.20 × 10^−5^), in classical and quasi-mesenchymal Collisson’s subtypes (*p* = 9.07 × 10^−4^), in basal-like Moffitt’s tumor subtype (*p* = 4.55 × 10^−4^), and in activated Moffitt’s stroma subtype (*p* = 4.69 × 10^−4^).

### 3.3. XPO1 Expression and Overall Survival

Overall survival data were available for 573 patients. With a median follow-up of 16 months (range, 1 to 156), 351 patients (61%) died, the two-year OS was 39% (95% confidence interval (CI), 35–44; Figure 2A), and the median OS was 19 months (range, 1 to 156). As shown in Figure 2B, XPO1 expression influenced OS, with 30% two-year OS (95% CI, 24–36) in the “XPO1-high” class versus 48% (95% CI, 42–56) in the “XPO1-low” class (*p* = 4.19 × 10^−5^, log-rank test). The respective median OS were 16 months (range: 1 to 126) versus 23 months (range: 1 to 156), and the hazard ratio (HR) for death was 1.56 (95% CI, 1.26–2.93) in the “XPO1-high” class when compared with the “XPO1-low” class (*p* = 4.78 × 10^−5^, Wald test).

In univariate analysis (Table 2), the other variables that were associated with OS (Wald test) were the AJCC stage (*p* = 3.98 × 10^−3^), the pathological lymph node involvement (*p* = 3.80 × 10^−5^), the Collisson’s classification (*p* = 7.62 × 10^−3^), the Moffitt’s tumor classification (*p* = 1.69 × 10^−5^), the Moffitt’s stroma classification (*p* = 4.33 × 10^−4^), and the Bailey’s classification (*p* = 1.26 × 10^−6^). Patients’ age and sex, and pathological tumor size and grade were not significantly associated with OS. In multivariate analysis, XPO1 expression remained significant when confronted with the significant clinicopathological variables (HR for death equal to 1.6 (95% CI, 1.23–2.09) in the “XPO1-high” class when compared with the “XPO1-low” class (*p* = 5.07 × 10^−4^, Wald test) and when confronted with the molecular subtypes (HR for death equal to 1.49 (95% CI, 1.18–1.87) in the “XPO1-high” class when compared with the “XPO1-low” class (*p* = 6.68 × 10^−4^, Wald test), suggesting an independent prognostic value. Due to the correlation between the four different molecular subtype classifications, we repeated the multivariate analyses by including XPO1 expression and each of the four classifications separately. As shown in Appendix A, XPO1 expression and each molecular classification remained significant. Of note, the same independent prognostic value was observed for XPO1 expression when analyzed in continuous value (respective p-values that are equal to 4.10 × 10^−4^ and 1.13 × 10^−4^).

### 3.4. XPO1 Expression and Associated Biological Processes

To explore the biological alterations that are associated with the *XPO1* expression status, we applied supervised analysis to the TCGA data set (*N* = 150). We identified 268 genes that were differentially expressed between the tumors with (*N* = 76) versus without (*N* = 74) *XPO1* upregulation, including 191 genes that were upregulated and 77 genes that were downregulated in the “*XPO1*-high” samples (Appendix A). Ontology analysis (Appendix A) revealed the strong involvement of genes that were overexpressed in the “*XPO1*-high” tumors in cell cycle, nuclear division, DNA repair, signal transduction, chromosome segregation, DNA replication, and RNA processing. Ontologies that were associated with the genes underexpressed in the “*XPO1*-high” tumors were fewer and mainly related to metabolism and development. The robustness of this gene signature was verified in the learning set, and more importantly confirmed in the independent validation set by using a metagene-based prediction score (Appendix A).

## 4. Discussion

The need for new therapeutic and/or prognostic targets is crucial in PAC. We have analyzed *XPO1* mRNA expression in 573 clinical PAC samples because of the promising therapeutic value of XPO1 inhibitors in oncology and the paucity of data in the literature: high expression was associated with shorter OS in multivariate analysis. To our knowledge, this is by far the largest study analyzing *XPO1* expression in PAC.

Our analysis was based on mRNA expression rather than protein expression as measured using immunohistochemistry (IHC) for several reasons: i) avoiding the limitations of IHC with different non-standardized protocols for XPO1; ii) working on an available large series of clinical samples; and, iii) searching for associations with expression of other genes on a whole-genome scale. When compared to normal pancreatic tissue, *XPO1* expression was higher in primary PAC. Of note, it was also higher in secondary tumors as compared to primary tumors. Expression was heterogeneous between samples in our series of 573 operated primary PAC. This range of expression values allowed for searching for correlations with clinicopathological features. Correlations existed with the pathological lymph node status (pN) and aggressive molecular subtypes, squamous Bailey’s subtype, quasi-mesenchymal Collisson’s subtype, basal-like Moffitt’s tumor subtype, and activated Moffitt’s stroma subtype. Such an association with adverse prognostic features was confirmed in univariate analysis with shorter metastasis-free survival (MFS) in the “*XPO1*-high” class. However, interestingly, such unfavorable prognostic value remained significant in multivariate analysis, suggesting independence. Our analysis was based on discrete values while using the mean expression level in normal tissues as cut-off, but a similar correlation was found when *XPO1* expression was analyzed as continuous values.

It is not surprising to find frequent high expression in PAC and association with poor prognosis given the XPO1 function of inactivation of TSPs. This has already been reported in several cancers [18,19,20,21,22,23,24,25,26,27,68,69]. Regarding PAC, to our knowledge, only three studies have described XPO1 protein expression in clinical samples using Western blot and IHC with different antibodies [23,52,53]. The first one, which was published in 2009 [23], included 69 primary pancreatic cancer samples and 10 normal tissues that were tested using Western blot. Increased XPO1 expression was shown in pancreatic cancer, and high expression was associated with increased serum levels of CEA and CA19-9, with tumor size, lymphadenopathy, and liver metastasis, and with shorter progression-free survival (PFS) and OS in uni- and multivariate analyses. The second study [53] concerned 91 pancreatic cancer tissues and 70 non-malignant pancreatic samples and it showed higher expression in cancer samples than in the matched normal control, but no correlation with the clinicopathological features and survival tested. In the last study [52], IHC was applied to a tissue microarray comprising 76 primary cancer samples: XPO1 was expressed in 86% of pancreatic cancers, and increased expression was correlated with both survivin expression and increased proliferative activity; no correlation with clinicopathological features and survival was searched.

Our analysis of genes upregulated in *XPO1*-high tumors identified several ontologies that were related to cell proliferation, such as cell cycle, nuclear division, chromosome segregation, and DNA replication; other ontologies, such as DNA repair, signal transduction, or RNA processing also agreed with the multiple protein targets of XPO1 exported from the cell nucleus to the cytoplasm in eukaryotic cells [13,16], and with recent publications showing that selinexor, which is an XPO1 inhibitor, reduces the expression of DNA damage repair proteins [51], and showing the correlation of XPO1 expression with the proliferative activity [52]. These correlations explain, at least in part, the oncogenic effect of XPO1 and the association with tumor stage (higher in metastatic samples than in primary tumors, and higher in primary tumors than in normal tissue), with shorter survival and with resistance to cytotoxic therapies. Importantly, they also provide indication that increased *XPO1* expression in PAC is likely associated with an increase in its biological activity.

## 5. Conclusions

In conclusion, we showed that *XPO1* mRNA expression is heterogeneous in PAC and is associated with progression stage and shorter survival independently from the other prognostic features. The strength of our study lies in the size of the series (the largest series of tumors reported to date regarding analysis of XPO1 expression), the biological and clinical relevance of *XPO1* expression, and its independent prognostic value. The limitations include its retrospective nature and associated biases, such as the lack of available information regarding the delivery or not of adjuvant chemotherapy for most of cases. No patient had received neoadjuvant chemotherapy, impeding the search for an eventual correlation of XPO1 expression with response to chemotherapy. Obviously, an analysis of larger series, retrospective, then prospective, is warranted to confirm our observation. Functional studies need to be conducted with fresh patient samples as well as retroactive meta-data analysis to link mRNA levels to actual protein expression and patient outcomes. If such a prognostic value is confirmed, *XPO1* expression might refine the prognostication and improve our ability to tailor adjuvant chemotherapy. However, more importantly, and given this unfavorable prognostic value and the likely association with increased XPO1 biological activity, patients with high level of *XPO1* expression would warrant a more aggressive treatment plan, which should include SINE compounds that are associated with classical drugs, notably the DNA-damaging agents. In the metastatic setting, clinical trials are ongoing, and it will be important to test whether *XPO1* mRNA expression can predict the clinical response to SINE compounds.

## Figures and Tables

**Figure 1 jcm-08-00596-f001:**
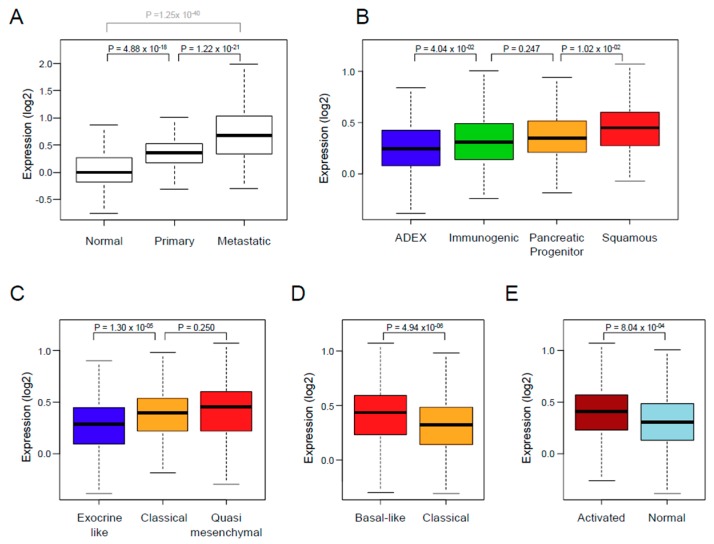
*XPO1* expression in PAC. (**A**) Box plots showing *XPO1* mRNA expression level (log_2_) in 95 normal pancreatic samples, 573 primary PAC, and 73 metastatic samples. For each box plot, median and ranges are indicated. The *p*-values are for Student *t*-test. (**B**–**E**) Similar to **A**, but in primary PAC only and according to the molecular subtypes that are defined by Bailey (**B**), Collisson (**C**), Moffitt (**D**: tumor subtypes), and Moffitt (**E**: stroma subtypes).

**Figure 2 jcm-08-00596-f002:**
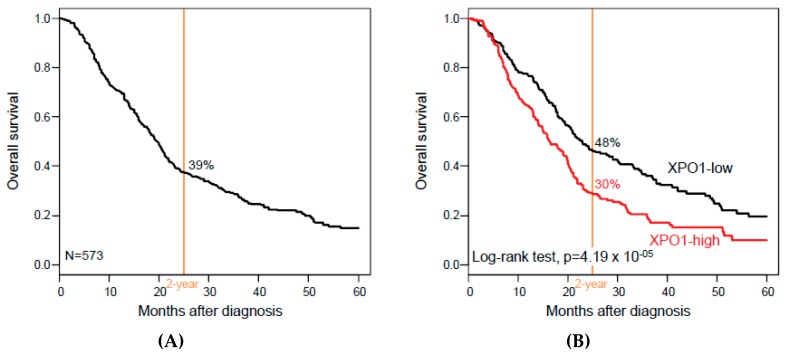
Overall survival (OS) in patients with PAC according to *XPO1* expression. (**A**) Kaplan–Meier metastasis-free survival (MFS) curves in all patients with PAC. (**B**) Similar to A, but according to the XPO1-based classification (“XPO1-high” and “XPO1-low” classes).

**Table 1 jcm-08-00596-t001:** Clinico-pathological and molecular characteristics of 573 primary pancreatic adenocarcinoma (PAC) samples in the whole population and in each exportin-1 (XPO1)-based group.

Charateristics	*N*	Global (*N* = 573)	*XPO1* ‘Low’ (*N* = 275)	*XPO1* ‘High’ (*N* = 298)	*p*-Value
Age at diagnosis (years)					0.348
≤60	106	106 (32%)	47 (29%)	59 (34%)	
>60	230	230 (68%)	116 (71%)	114 (66%)	
Sex					0.159
female	159	159 (47%)	83 (51%)	76 (43%)	
male	180	180 (53%)	80 (49%)	100 (57%)	
AJCC Stage					0.507
1	53	53 (11%)	30 (13%)	23 (9%)	
2	413	413 (85%)	191 (83%)	222 (86%)	
3	10	10 (2%)	4 (2%)	6 (2%)	
4	12	12 (2%)	6 (3%)	6 (2%)	
Pathological type					0.853
ductal	401	401 (93%)	196 (92%)	205 (93%)	
other	31	31 (7%)	16 (8%)	15 (7%)	
Pathological grade					0.179
1	23	23 (9%)	13 (10%)	10 (8%)	
2	144	144 (57%)	78 (61%)	66 (52%)	
3	85	85 (33%)	37 (29%)	48 (38%)	
4	2	2 (1%)	0 (0%)	2 (2%)	
Pathological tumor size (pT)					0.538
pT1	16	16 (4%)	8 (5%)	8 (4%)	
pT2	54	54 (15%)	30 (17%)	24 (12%)	
pT3	285	285 (78%)	132 (76%)	153 (80%)	
pT4	11	11 (3%)	4 (2%)	7 (4%)	
Pathological lymph node status (pN)					**1.97 × 10^−2^**
negative	128	128 (30%)	73 (36%)	55 (25%)	
positive	296	296 (70%)	131 (64%)	165 (75%)	
Collisson subtypes					**9.07 × 10^−4^**
classical	257	257 (45%)	114 (41%)	143 (48%)	
exocrine-like		199 (35%)	116 (42%)	83 (28%)	
quasi-mesenchymal	117	117 (20%)	45 (16%)	72 (24%)	
Moffit subtypes, ‘tumor’					**4.55 × 10^−4^**
basal-like	229	229 (40%)	89 (32%)	140 (47%)	
classical	344	344 (60%)	186 (68%)	158 (53%)	
Moffit subtypes, ‘stroma’					**4.69 × 10^−4^**
Activated	324	324 (59%)	132 (51%)	192 (66%)	
Normal	222	222 (41%)	125 (49%)	97 (34%)	
Bailey subtypes					**3.20 × 10^−5^**
ADEX	124	124 (22%)	77 (28%)	47 (16%)	
immunogenic	99	99 (17%)	53 (19%)	46 (15%)	
pancreatic progenitor	141	141 (25%)	70 (25%)	71 (24%)	
squamous	209	209 (36%)	75 (27%)	134 (45%)	
Follow-up, months (range)	573	16 (1-156)	17 (1-156)	13 (1-126)	0.08
2-year OS (95% CI)	573	39% (35–44)	48% (42–56)	30% (24–36)	**4.19 × 10^−5^**
Median OS, months (range)	573	19 (1–156)	23 (1–156)	16 (1–126)	**1.67 × 10^−3^**

AJCC: American Joint Committee on Cancer. OS: Overall survival. CI: confidence interval.

**Table 2 jcm-08-00596-t002:** Uni- and multivariate prognostic analyses for OS.

Characteristics		Univariate			Multivariate			Multivariate		
		*N*	HR (95% CI)	*p*-Value	*N*	HR (95% CI)	*p*-Value	*N*	HR (95% CI)	*p*-Value
Age at diagnosis (years)	>60 vs. ≤60	336	1.14 (0.83–1.56)	0.410						
Sex	male vs. female	339	1.12 (0.85–1.49)	0.421						
AJCC Stage	2 vs. 1	488	2.00 (1.32–3.02)	**3.98 × 10^−3^**	417	1.46 (0.83–2.55)	0.188			
	3 vs. 1		2.95 (1.27–6.83)		417	1.46 (0.54–3.94)	0.454			
	4 vs. 1		2.96 (1.27–6.90)		417	0.98 (0.13–7.37)	0.983			
Pathological type	other vs. ductal	432	0.94 (0.54–1.64)	0.822						
Pathological grade	2 vs. 1	254	1.69 (0.67–4.23)	0.056						
	3 vs. 1		2.58 (1.02–6.51)							
	4 vs. 1		2.92 (0.56–15.2)							
Pathological tumor size (pT)	2 vs. 1	366	1.92 (0.80–4.61)	0.097						
	3 vs. 1		2.35 (1.04–5.32)							
	4 vs. 1		3.40 (1.18–9.82)							
Pathological lymph node status (pN)	1 vs. 0	424	1.85 (1.38–2.48)	**3.80 × 10^−5^**	417	1.51 (1.07–2.13)	**1.87 × 10^–2^**			
Collisson subtypes	exocrine-like vs. classical	573	0.99 (0.77–1.26)	**7.62 × 10^−3^**				546	1.00 (0.71–1.41)	0.997
	quasi-mesenchymal vs. classical		1.47 (1.12–1.91)					546	0.93 (0.67–1.30)	0.674
Moffit subtypes, ‘tumor’	classical vs. basal-like	573	0.63 (0.51–0.77)	**1.69 × 10^−5^**				546	1.01 (0.74–1.38)	0.961
Moffit subtypes, ‘stroma’	normal vs. activated	546	0.67 (0.53–0.84)	**4.33 × 10^−4^**				546	0.79 (0.62–1.01)	0.060
Bailey subtypes	immunogenic vs. ADEX	573	0.89 (0.62–1.26)	**1.26 × 10^−6^**				546	0.87 (0.55–1.38)	0.566
	pancreatic progenitor vs. ADEX		0.98 (0.71–1.35)					546	0.95 (0.62–1.45)	0.805
	squamous vs. ADEX		1.74 (1.31–2.32)					546	1.57 (1.02–2.43)	**4.24 × 10^−2^**
XPO1	high vs. low	573	1.56 (1.26–1.93)	**4.78 × 10^−5^**	417	1.6 (1.23–2.09)	**5.07 × 10^−4^**	546	1.49 (1.18–1.87)	**6.68 × 10^−4^**

HR: hazard ratio.

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
