# Peer review of "XPO1 Expression Is a Poor-Prognosis Marker in Pancreatic Adenocarcinoma"

_jcm, 2019, doi:10.3390/jcm8050596_

Round 1
Reviewer 1 Report
The manuscript is well structured, nicely performed and provides new evidence on the correlation between XPO1 expression and survival in pancreatic adenocarcinoma patients. The study contributes to the efforts of identifying the prognostic value of XPO1 in pancreatic cancer.
The study is retrospective and may have a bias as indicated by the authors in the conclusion section.
It was not clear for me if any of the samples included in the study were from patients treated with chemotherapeutic agents (neo-adjuvant, adjuvant etc). If yes, can authors comment on if the XPO1 expression has any correlation with chemotherapy treatment response?
Reviewer 1 Report
The manuscript is well structured, nicely performed and provides new evidence on the correlation between XPO1 expression and survival in pancreatic adenocarcinoma patients. The study contributes to the efforts of identifying the prognostic value of XPO1 in pancreatic cancer.
The study is retrospective and may have a bias as indicated by the authors in the conclusion section.
It was not clear for me if any of the samples included in the study were from patients treated with chemotherapeutic agents (neo-adjuvant, adjuvant etc). If yes, can authors comment on if the XPO1 expression has any correlation with chemotherapy treatment response?
Author Response
Dear Editor-in-Chief,
Please find enclosed a revised version of the manuscript « jcm-486984 » by Birnbaum and colleagues, entitled « XPO1 expression is a poor-prognosis marker in pancreatic adenocarcinoma » that we had submitting for publication as an original article in “Journal of Clinical Medicine” for a special issue on “Current Standards and New Innovative Approaches for Treatment of Pancreatic Cancer” (Dead line: June 2019).
We thank the two Reviewers for their positive and helpful comments, which have been taken into account as follows.
Reviewer 1’s comments and Authors’ responses
The manuscript is well structured, nicely performed and provides new evidence on the correlation between XPO1 expression and survival in pancreatic adenocarcinoma patients. The study contributes to the efforts of identifying the prognostic value of XPO1 in pancreatic cancer.
We thank the reviewer for his/her positive comment.
1. The study is retrospective and may have a bias as indicated by the authors in the conclusion section. It was not clear for me if any of the samples included in the study were from patients treated with chemotherapeutic agents (neo-adjuvant, adjuvant etc). If yes, can authors comment on if the XPO1 expression has any correlation with chemotherapy treatment response?
On line 155, we have added the following sentence: «None of them had received neoadjuvant chemotherapy or radiotherapy.”
Information about the delivery of chemotherapy was missing for most of patients and represents a classical bias of retrospective studies. On line 276, the sentence “Limitations include its retrospective nature and associated biases.” has been replaced by the following sentence: “Limitations include its retrospective nature and associated biases, such as the lack of available information about the delivery or not of adjuvant chemotherapy for most of cases. No patient had received neoadjuvant chemotherapy, impeding to search for an eventual correlation of XPO1 expression with response to chemotherapy.”

Reviewer 2 Report
Birnbaum et al. show that high expression of the karyopherin protein, exportin 1 (XPO1) transcripts negatively correlates with survival in pancreatic adenocarcinoma (PAC) patients. They analyzed the gene expression data from 741 clinical pancreatic samples and found that age, gender and tumor stage did not impact overall survival. Molecular subtype, tumor recurrence (metastasis) and lymph node involvement influenced on overall survival without diminishing the impact of XPO1 on survival. Multivariate as well as continuous value analysis of the expression levels of XPO1 in conjunction with molecular subtypes suggests that XPO expression by itself is a valuable prognostic indicator of survival. Their results indicate that knowledge of the XPO1 status of patients can help determine who will best benefit from XPO1 inhibitors currently being tested in phase 1 trials to improve the marginal survival rates obtained with standard of care treatment.
1. Line 42, “through” instead of though
2. Figure 1 legend (A) “Box” in place of Bot
3. The authors analyzed the effect of the different classes of PAC subtypes on XPO1 correlation to survival. They did not comment on the overlap of subtypes and how this affected the survival correlation of XPO1 when all classes assigned to a particular sample were taken into consideration compared to the individual subtype correlation analysis (i.e. is XPO1 independent of tumor subtype designation or specific to a particular molecular signature that can be found across tumor samples).
The gene ontologies of genes overexpressed in association with elevated XPO1 mRNA levels suggest dysregulation of tumor suppressor genes and other cellular processes that promote tumor progression. However, functional studies will need to be conducted in the future with fresh patient samples as well as retroactive meta-data analysis to link mRNA levels to actual protein expression and patient outcomes.
This study by Birnbaum et al. is an important addition to the field and has a translational application in the clinic. Knowing the XPO1 status of PAC patients will help personalize treatment regimens, assigning PAC patients to clinical trials as well as improving the overall survival as we move towards a precision medicine approach. This will be especially true for patients with secondary tumors that make up most PAC patients.
Reviewer 2’s comments and Authors’ responses
Dear Editor-in-Chief,
Please find enclosed a revised version of the manuscript « jcm-486984 » by Birnbaum and colleagues, entitled « XPO1 expression is a poor-prognosis marker in pancreatic adenocarcinoma » that we had submitting for publication as an original article in “Journal of Clinical Medicine” for a special issue on “Current Standards and New Innovative Approaches for Treatment of Pancreatic Cancer” (Dead line: June 2019).
We thank the two Reviewers for their positive and helpful comments, which have been taken into account as follows.
Birnbaum et al. show that high expression of the karyopherin protein, exportin 1 (XPO1) transcripts negatively correlates with survival in pancreatic adenocarcinoma (PAC) patients. They analyzed the gene expression data from 741 clinical pancreatic samples and found that age, gender and tumor stage did not impact overall survival. Molecular subtype, tumor recurrence (metastasis) and lymph node involvement influenced on overall survival without diminishing the impact of XPO1 on survival. Multivariate as well as continuous value analysis of the expression levels of XPO1 in conjunction with molecular subtypes suggests that XPO expression by itself is a valuable prognostic indicator of survival. Their results indicate that knowledge of the XPO1 status of patients can help determine who will best benefit from XPO1 inhibitors currently being tested in phase 1 trials to improve the marginal survival rates obtained with standard of care treatment.
1. Line 42, “through” instead of though
On line 42, « though » has been replaced by « through ».
2. Figure 1 legend (A) “Box” in place of Bot
On line 177, « Bot » has been replaced by « Box ».
3. The authors analyzed the effect of the different classes of PAC subtypes on XPO1 correlation to survival. They did not comment on the overlap of subtypes and how this affected the survival correlation of XPO1 when all classes assigned to a particular sample were taken into consideration compared to the individual subtype correlation analysis (i.e. is XPO1 independent of tumor subtype designation or specific to a particular molecular signature that can be found across tumor samples).
We thank the reviewer for this comment. Indeed, because of the correlation between the four different molecular subtype classifications, we repeated the multivariate analyses by including XPO1 expression and each of the four classifications alone and separately. The results are shown in a new Supplementary Table (Supplementary Table 2) and described in the Results section.
On line 204, we have added the following sentence: “Because of the correlation between the four different molecular subtype classifications, we repeated the multivariate analyses by including XPO1 expression and each of the four classifications separately. As shown in Supplementary Table 2, XPO1 expression and each molecular classification remained significant.”
The initial Supplementary Tables 2 and 3 have been renumbered Supplementary Tables 3 and 4 respectively.
4. The gene ontologies of genes overexpressed in association with elevated XPO1 mRNA levels suggest dysregulation of tumor suppressor genes and other cellular processes that promote tumor progression. However, functional studies will need to be conducted in the future with fresh patient samples as well as retroactive meta-data analysis to link mRNA levels to actual protein expression and patient outcomes.
We agree. In the Conclusion, on line 281, we have added the following sentence : « And, functional studies need to be conducted with fresh patient samples as well as retroactive meta-data analysis to link mRNA levels to actual protein expression and patient outcomes. »
5. This study by Birnbaum et al. is an important addition to the field and has a translational application in the clinic. Knowing the XPO1 status of PAC patients will help personalize treatment regimens, assigning PAC patients to clinical trials as well as improving the overall survival as we move towards a precision medicine approach. This will be especially true for patients with secondary tumors that make up most PAC patients.
We thank the reviewer for his/her positive comment.
As you can see, we have answered all questions raised by the two Reviewers and modified the manuscript as suggested. We hope that this improved version will meet with your approval for publication in your journal.
Thank you very much for your consideration and help in the process.
Sincerely yours,
Dr David Birnbaum, MD PhD – Pr François Bertucci, MD PhD
